

# Optimizing co-location calibration periods for low-cost sensors

Misti Levy Zamora[1,2,3*], Colby Buehler[3,4], Abhirup Datta[5], Drew R. Gentner[3,4], Kirsten Koehler[2,3]

[1]University of Connecticut Health Center, Department of Public Health Sciences UConn School of Medicine, 263
Farmington Avenue, Farmington, CT, USA 06032-1941
[2]Johns Hopkins University Bloomberg School of Public Health, Environmental Health and Engineering
615 N Wolfe St, Baltimore, MD, USA 21205-2103
[3]SEARCH (Solutions for Energy, Air, Climate and Health) Center, Yale University, New Haven, CT, USA 06520
[4]Yale University, Chemical and Environmental Engineering, PO Box 208286, New Haven, CT, USA 06520
[5]Johns Hopkins University Bloomberg School of Public Health, Department of Biostatistics
615 N. Wolfe Street, Baltimore, MD, USA 21205-2103

*Correspondence to*: Misti Levy Zamora (mzamora@UCHC.edu)

**Abstract.** Low-cost sensors are often co-located with reference instruments to assess their performance and establish calibration equations, but limited discussion has focused on whether the duration of this calibration period can be optimized. We placed a multipollutant monitor that contained sensors that measure particulate matter smaller than 2.5 mm ($PM_{2.5}$), carbon monoxide (CO), nitrogen dioxide ($NO_2$), ozone ($O_3$), and nitric oxide (NO) at a reference field site for one year. We developed calibration equations using randomly co-location subsets spanning 1 to 180 consecutive days out of the 1-year period and compared the potential root mean square errors (RMSE) and Pearson correlation coefficients (r). The co-located calibration period required to obtain consistent results varied by sensor type, and several factors increased the co-location duration required for accurate calibration, including the response of a sensor to environmental factors, such as temperature or relative humidity (RH), or cross-sensitivities to other pollutants. Using measurements from Baltimore, MD, where a broad range of environmental conditions may be observed over a given year, we found diminishing improvements in the median RMSE for calibration periods longer than about six weeks for all the sensors. The best performing calibration periods were the ones that contained a range of environmental conditions similar to those encountered during the evaluation period (i.e., all other days of the year not used in the calibration). With optimal, varying conditions it was possible to obtain an accurate calibration in as little as one week for all sensors, suggesting that co-location can be minimized if the period is



strategically selected and monitored so that the calibration period is representative of the desired measurement setting.

## 1. Introduction

Instrument calibration is one of the main processes used to ensure instrument accuracy. In one method of calibration, measurements are compared between an uncalibrated instrument and a reference

instrument, which can then be used to adjust the output of the uncalibrated instrument to see whether the data can meet performance standards (often in terms of accuracy and precision). In the case of low-cost air-pollution sensors, the raw output is often a voltage or resistance instead of a concentration, so a calibration curve is needed to convert the raw output into practical units. Cross-sensitivities to environmental conditions or other pollutants, non-linear responses, and variability between sensor

units are common difficulties that must be considered when working with low-cost sensor data (Van Zoest et al., 2019; Levy Zamora, 2022; Li et al., 2021; Spinelle et al., 2015; Ripoll et al., 2019). Several methodologies have been used to derive the calibration equations needed to convert the raw data into useable concentrations, such as exposing the sensors to known concentrations in a laboratory setting and co-locating the sensors with a reference instrument, often in a similar setting to which the sensor

is to be used (Taylor, 2016; Zimmerman et al., 2018; Mead et al., 2013; Ikram et al., 2012; Hagler et al., 2018; Cross et al., 2017; Holstius et al., 2014; Mukherjee et al., 2019; Gao et al., 2015; Heimann et al., 2015; (Scaqmd), 2016a, 2017, 2016b; Levy Zamora et al., 2018a). Field co-location is a widely used calibration method, but a tradeoff must be made between the time dedicated to collecting calibration data and the data collected at the final measurement location. There is currently no

standardized co-location duration, and the reported co-location durations for low-cost sensors with reference instruments in recent work have varied from several days to several months (Mukherjee et al., 2019; Gao et al., 2015; Topalović et al., 2019; Kim et al., 2018; (Aq-Spec), 2018; Spinelle et al., 2017; Pinto et al., 2014; Datta et al., 2020). To date, little discussion has focused on whether the





selected periods were adequate for the deployment period or whether the calibration period can be
optimized in future studies (Topalović et al., 2019; Okorn and Hannigan, 2021). In one study that
assessed the impacts of co-location duration for a low-cost sensor, Okorn et al (Okorn and Hannigan,
2021) randomly selected calibration periods up to six weeks in duration from six weeks of methane
data in Los Angeles. The calibration equations were then applied to data from an earlier month in the
same location. They reported that longer calibration periods produced fits with lower bias than fits
from shorter calibration periods. In that study, the one-week calibrations yielded the best $R^2$ values.

The central goal of this specific work was to identify the key factors that influence the duration of
a co-location required to obtain sufficient data to achieve consistent calibrate curves for five low-cost
sensors (particulate matter smaller than 2.5 microns ($PM_{2.5}$), carbon monoxide (CO), ozone ($O_3$),
nitrogen dioxide ($NO_2$), and nitrogen monoxide (NO)) (Buehler et al., 2021). In addition, we aim to
identify how this necessary calibration period can be optimized.

## 2. Methods

### 2.1. Data Collection

Data collected at two sites were used in the co-location analyses based on the availability of
reference instrumentation. The CO (Alphasense CO-A4 sensor), $NO_2$ (Alphasense NO2-A43F), NO
(Alphasense NO-A4), and $O_3$ (MiCS-2614) sensors were co-located with reference instruments at the
Maryland Department of the Environment's (MDE) Essex site (ID = 240053001) in Baltimore County,
Maryland. The $PM_{2.5}$ sensor (Plantower PMS A003) was concurrently co-located with a reference
instrument at the MDE Oldtown site (ID = 245100040) in Baltimore City, Maryland. The Essex site
(39.310833, -76.474444) is about 11 km east of the Oldtown site (39.298056, −76.604722). Additional
details about the sensors in the multipollutant monitor have been described in detail by Buehler et al.
(Buehler et al., 2021) and Levy Zamora et al. (Levy Zamora, 2022). Co-location data from February
1, 2019, to February 1, 2020, were used in the $PM_{2.5}$ analysis, and co-location data from February 1 to





December 20, 2019, were used in the CO, NO, $NO_2$, and $O_3$ sensors analyses. Due to an issue affecting the gas sensor inlet on the Essex monitor, the $O_3$, $NO_2$, and NO sensor data were unavailable after
December 20, 2019. Hourly average data was used in all analyses. Both reference sites also measured hourly averaged temperature and relative humidity (RH). The ambient temperature and RH ranged between -11 and 36ºC and 14 and 95% over the full year, respectively. The temperatures and RHs measured inside the multipollutant pollutant monitors were slightly different from the ambient values due to direct sunlight warming the monitors and the small amount of heat produced by the sensors
themselves within the box. The box temperatures and RHs ranged between -8 and 45 ºC and 14 and 80%, respectively.

### 2.2. Assessing the Role of Co-location Duration

We use different subsets of the full co-location period to create a suite of hypothetical co-location durations based on which the calibration models will be trained. For each hypothetical calibration co-
location duration (i.e., ranging from 1 to180 consecutive days), 250 sample calibration test periods were randomly selected each of which had that total duration. These test periods were used in the sensitivity analysis at each test condition to assess the range of potential resulting root mean squared error (RMSE) values and Pearson correlation coefficients (r). For example, a calibration duration of 1 day indicates that a 24-hour period was randomly selected out of the available data, referred to as the
"calibration period", and the data from those 24 hours was used to develop the calibration equations (see below) relating the raw sensor data to ambient conditions. This was then evaluated against all days not included in the calibration period, referred to as the "evaluation period". The randomly chosen calibration periods could overlap, but no two periods were exactly the same. In Supplemental Figure 1, the start times of 250 randomly selected $PM_{2.5}$ calibrations are shown as an example. Each tested
co-location duration produced 250 RMSE and r values, and only calibration periods with at least 70% valid sensor and reference data were used in the analyses (e.g., a 24 hr calibration period needed to have more than 16 hours of valid data for both instruments). No laboratory or manufacturers



information was used to additionally calibrate the sensors in this work. All data analysis was conducted using Matlab 2020a.

Sensor data from the calibration period was used to determine the coefficients for multiple linear regression (MLR) models based on previously identified known environmental factors influencing concentration for each sensor (Levy Zamora, 2022). A generic MLR model is given by:

$$Reference_{Pollutant}(t) = \beta_o + \beta_1 * Sensor_{Pollutant}(t) + \sum_1^n \beta_n * Predictor_n(t) \quad (1)$$

where $Reference_{pollutant}$ is the reference concentration at time t for a given pollutant, $\beta_0$ is the constant intercept, $\beta_1$ is the coefficient applied to the uncalibrated $Sensor_{pollutant}$ value for a given pollutant at time t, and $\beta_n$ is the coefficient applied to $Predictor_n$. Levy Zamora et al. (Levy Zamora, 2022) have reported the predictors needed to calibrate these five low-cost sensors in detail. Briefly, the $PM_{2.5}$ sensor model incorporated temperature and RH as predictors. The CO sensor model included

temperature, RH, and time. The $NO_2$ sensor model included temperature, RH, NO, $O_3$, and time. The $O_3$ model included temperature, RH, $NO_2$, and time, and the NO model included temperature and CO as predictors. The CO, $O_3$, and $NO_2$ sensors may exhibit baseline drift over the year, which is why the time predictors were included. The data used as the predictors came from the other sensors in the multipollutant monitor (e.g., the NO sensor model used the co-located low-cost CO sensor for the CO

predictor). Once the regression coefficients were determined for a calibration period, this equation was applied to all data in the corresponding evaluation period.

For each calibration period tested, the RMSE and correlation coefficients were determined by comparing the 1-hour averaged reference and corrected sensor data from all hours during the evaluation period. The RMSE was calculated using Equation 2 where $Reference_i$ and

$Predicted_i$ are the corresponding i-th 1-hour averaged concentrations from the evaluation period with N data points.

$$\text{RMSE} = \sqrt{\frac{\sum_{i=1}^{N}(Reference_i - Predicted_i)^2}{N}} \quad (2)$$





An RMSE value of 0 would indicate a perfect agreement between the reference and sensor. The correlation coefficient is a measure of the linear correlation between two data sets. It is a value between −1 and 1, where 1 indicates a strong positive relationship, -1 indicates a strong negative relationship, and 0 has no discernible relationship. The median RMSE and median r referenced in this manuscript refer to the median value from all the 250 calibration scenarios for each duration. Outliers are defined as a value that is more than three scaled median absolute deviations (MAD) away from the median.

We hypothesize that a user could strategically choose a co-location period to minimize the calibration period and that co-location duration is not the only factor to consider when optimizing co-locating an instrument for calibration. In these analyses, we use the term "coverage" to indicate the representativeness of environmental conditions during a calibration period compared to that observed across the full data set (calibration and evaluation periods). In order to visualize how the environmental conditions during the calibration period compared to the evaluation period, we compared the range of temperature, RH, and other key pollutants from each period. For example, if the full RH ranged between 10 and 90% and the calibration period ranged between 20 and 60%, the RH coverage of that calibration period would be 50% (40/80).

$$Coverage = \frac{Maximum\ Value_{Calibration\ Period} - Minimum\ Value_{Calibration\ Period}}{Maximum\ Value_{Full\ Year} - Minimum\ Value_{Full\ Year}} \text{ X } 100 \quad (3)$$

## 3. Results and Discussion

### 3.1. Impact of colocation duration on calibration performance

The range of RMSE values from 250 calibration periods in the sensitivity analysis of six co-location durations (i.e., 1 day, 1 week, 1 month, 6 weeks, 3 months, and 6 months) for all five low-cost sensors are shown in Table 1, and the box plots of the RMSEs from co-location durations ranging 1-180 days are shown in Figures 1 ($PM_{2.5}$ and CO) and 2 ($NO_2$, $O_3$, and NO). Overall, longer



calibrations resulted in lower median RMSE values. The greatest improvements for median RMSE values were observed when increasing the co-location duration from 1 day to about two weeks. After about six weeks, diminishing improvements were observed in the median RMSEs for all the sensors except ozone. The median RMSE for ozone decreased by about 5 ppb when increasing the duration

from 6 weeks to 6 months. There were also a limited number of high outlier RMSEs for any of the sensors after about two months indicating that most of the 250 calibrations were generally yielding similar RMSEs. In addition, the lowest RMSE values (e.g., 1$^{st}$ percentile) were similar for all co-location durations longer than about one week for many of the sensors. This suggests that optimized calibration periods can yield high-performance calibrations. For example, the RMSEs from the 1-week

calibration periods for the $PM_{2.5}$ sensor ranged between 3.1 and 18.3 µg/m$^3$, and the 6-month calibrations ranged between 3.2 and 3.7 µg/m$^3$. The 1$^{st}$ percentile RMSEs for the 1 week and 6 months were also similar for CO (61 and 51 ppb, respectively), $NO_2$ (4.1 and 3.6 ppb, respectively), $O_3$ (9.1 and 8.1 ppb, respectively), and NO (3.3 and 2.9 ppb, respectively). The 10$^{th}$ percentile RMSEs were similar after about 1 month for most sensors. For example, the 10$^{th}$ percentile for PM was 3.4 at 1

month and 3.5 µg/m$^3$ at 6 months (CO: 66 and 69 ppb, respectively; $NO_2$: 4.3 and 4.1 ppb, respectively; $O_3$: 11.0 and 8.4 ppb, respectively; NO: 3.5 and 2.9 ppb, respectively). The differences between the 1$^{st}$ and 99$^{th}$ percentile RMSE for the 6-month scenarios were comparatively small for all sensors compared to the overall concentrations and ranges (e.g., the RMSE range at 6 months for $PM_{2.5}$ was 0.5 µg/m$^3$ compared to the annual average concentration of 8.3 µg/m$^3$).

The ranges of correlation coefficients for the five low-cost sensors are shown in Table 2, and the box plots of the r values from co-location durations between 1-180 days are shown in Figure 1 ($PM_{2.5}$ and CO) and Supplemental Figure 2 ($NO_2$, $O_3$, and NO). Overall, longer calibrations also resulted in higher r values, though in some individual test periods it was possible to produce correlation coefficients at or above 0.6 in as little as 1 day for all five sensors. After about six weeks, only

incremental improvements were observed in the median correlations for all the sensors. For example, the greatest improvement in the median correlation after 6-weeks was observed for ozone which



increased from 0.71 at 6-weeks to 0.84 at 6-months. All of the sensors were able to achieve reliably high correlations without poorly-performing outlier cases (e.g., all 250 calibrations produced r > 0.6), but the co-location durations required to reduce this risk of outliers ranged between 18 days for the

NO sensor and about 120 days for the CO sensor (Figure 1, Supplemental Figure 2).

|  | 1 Day | 1 Week | 1 Month | 6 Weeks | 3 Months | 6 Months |
|---|---|---|---|---|---|---|
| **PM$_{2.5}$ (µg/m³)** | 44.9 (5.2 − 400) | 6.6 (3.1 − 18.3) | 3.4 (3.1 − 9.1) | 3.4 (3.2 − 7.9) | 3.5 (3.2 − 5.6) | 3.6 (3.2 − 3.7) |
| **CO (ppb)** | 4870 (196 − 28,580) | 437 (61 − 1,630) | 125 (57 − 231) | 98 (59 − 219) | 77 (57 − 135) | 76 (51 − 105) |
| **NO$_2$ (ppb)** | 22.4 (7.8 − 1830) | 8.6 (4.1 − 21.8) | 6.1 (4.1 − 10.5) | 6.1 (3.9 − 8.7) | 6.0 (3.7 − 7.8) | 4.9 (3.6 − 7.6) |
| **O$_3$ (ppb)** | 721.2 (15.2 − 10,100) | 50.8 (9.1 − 267.8) | 15.7 (8.9 − 27.1) | 15.8 (8.2 − 22.8) | 15.0 (8.4 − 23.0) | 10.3 (8.1 − 12.6) |
| **NO (ppb)** | 16.3 (4.2 − 624) | 7.5 (3.5 − 72.4) | 4.3 (3.3 − 6.2) | 3.5 (3.1 − 4.7) | 3.6 (2.4 − 4.1) | 3.2 (2.9 − 3.6) |

**Table 1. The median and range (1st to 99th percentile) of RMSE from 250 calibration runs from six co-location lengths (1 day, 1 week, 6 weeks, 1 month, 3 months, and 6 months) for five low-cost sensors. The median and range (min to max) of PM2.5, CO, NO2, O3, and NO reference concentrations were 7 (1-53) µg/m³, 199 (100 -2950) ppb, 5.5 (1-58) ppb, 32 (1-110) ppb, and 0.5 (0.1-136.5) ppb, respectively.**

|  | 1 Day | 1 Week | 1 Month | 6 Weeks | 3 Months | 6 Months |
|---|---|---|---|---|---|---|
| **PM$_{2.5}$** | 0.11 (-0.78 − 0.70) | 0.66 (-0.61 − 0.80) | 0.77 (0.57 − 0.82) | 0.79 (0.66 − 0.82) | 0.80 (0.69 − 0.83) | 0.84 (0.78 − 0.87) |
| **CO** | 0.18 (-0.48 − 0.73) | 0.41 (-0.40 − 0.90) | 0.76 (-0.21 − 0.92) | 0.86 (-0.17 − 0.92) | 0.88 (0.54 -0.92) | 0.92 (0.88 − 0.95) |
| **NO$_2$** | 0.49 (-0.58 − 0.82) | 0.70 (0.39 − 0.88) | 0.75 (0.63 − 0.89) | 0.77 (0.69 − 0.88) | 0.78 (0.74 − 0.88) | 0.85 (0.76 − 0.88) |
| **O$_3$** | 0.07 (-0.47 − 0.63) | 0.30 (-0.18 − 0.88) | 0.70 (0.17 − 0.90) | 0.71 (0.36 − 0.91) | 0.74 (0.61 − 0.92) | 0.84 (0.81 − 0.90) |
| **NO** | 0.27 (-0.89 − 0.95) | 0.88 (-0.23 − 0.95) | 0.94 (0.73 − 0.96) | 0.94 (0.86 − 0.96) | 0.95 (0.94 − 0.97) | 0.97 (0.97 − 0.98) |

**Table 2. The median and range (1st to 99th percentile) of correlation coefficients (r) from 250 calibration runs from six co-location**

**lengths (1 day, 1 week, 1 month, 6 weeks, 3 months, and 6 months) for five low-cost sensors.**







**Figure 1. The potential range of A-B) RMSE and C-D) correlation coefficients (r) for a given co-location length for the low-cost PM2.5 and CO sensors. A calibration length of 1 day indicates that a random, continuous 24-hour period was selected out of all available days. The RMSE for a given sample calibration was determined by comparing the 1-hour averaged reference and corrected sensor data from the days during the evaluation period (i.e., all other days of the year not used in the calibration). For each calibration length tested, 250 sample calibration periods were used to assess the range of potential RMSE and correlation coefficients. All sensors were calibrated using previously identified predictors in a multiple linear regression using data from the calibration period only. Reference PM2.5 concentrations ranged between 1 and 53 µg/m3, with a median concentration of 7 µg/m3, and reference CO concentrations ranged between 100 and 2947 ppb, with a median concentration of 199 ppb.**




**Figure 2. The potential range of RMSE values for a given co-location length for three low-cost sensors (NO₂, O₃, and NO). A calibration length of 1 day indicates that a random 24-hour period was selected out of all available days between February 2019 and February 2020. The RMSE for a given test calibration period was determined by comparing the 1-hour averaged reference and the corrected sensor data associated with that calibration across the evaluation period (all days not included in the calibration period). For each calibration length, 250 randomly selected calibration periods were used to assess the potential RMSE range. All sensors were calibrated using previously identified predictors in a multiple linear regression using data from the calibration period only. The reference NO₂ concentrations ranged between 1 and 58 ppb over the full year, with a median concentration of 5 ppb. The reference O₃ concentrations ranged between 1 and 110 ppb, with a median concentration of 31 ppb. The reference NO concentrations ranged between 0.1 and 137 ppb, with a median concentration of 0.5 ppb.**





### 3.2. Selecting Optimal Calibration Conditions for Co-location Periods

The results show that the calibration performance from shorter-term co-locations varies considerably depending on the chosen co-location period. If a user wanted all 250 potential co-location periods for the PM$_{2.5}$ sensor to have an RMSE below 4 µg/m$^3$ and an r > 0.6, the minimum co-location duration that would ensure all calibration periods satisfied these two requirements would be 108 days at this site. However, 22% of the 7-day co-locations also produced calibrations that satisfied these two

requirements, so we analyzed the environmental factors during one-week calibrations that led to low and high RMSEs. In Figure 3 and Supplemental Figure 3, results from two one-week calibration periods are shown to demonstrate the range of potential RMSE values for the PM$_{2.5}$ sensor with differences in calibration conditions. The corresponding raw sensor, temperature, and RH data are also shown in the lower panels of Figure 3. In this comparative example, "Calibration Period 1" produced

more accurate concentrations during the evaluation periods (RMSE = 3.1 µg/m$^3$), whereas "Calibration Period 2" performed poorly (RMSE = 19.5 µg/m$^3$). Calibration Period 1 included a wider range of concentrations (1-45 µg/m$^3$), temperatures (-2 - 12ºC), and RHs (17-93%) and was able to yield similar concentrations as the reference data for the full year, whereas Calibration Period 2 was more limited in its range of conditions (6-37 µg/m$^3$, 21-30 ºC, and 42-88%, respectively) and performed reasonably

only during the summer months. In addition, the largest 6-month RMSE (e.g., 3.7 µg/m$^3$ for PM2.5 and 12.6 ppb for Ozone; Table 1) were generally comprised of more months when ambient concentrations were low and less variable (summer and winter, respectively), and the scenarios with the lowest RMSE included the months with the greatest concentrations observed in the data set.

        Based on these results, we hypothesized that a key element governing good calibration outcomes

is if the calibration co-location period is representative of the evaluation period in terms of the required predictors in Equation 1. Note, the required predictors are distinct for each sensor type, so optimal periods may differ by sensor. To evaluate this hypothesis, the median RMSEs for three sensors (PM$_{2.5}$, NO$_2$, and CO) were plotted as a function of the coverage of key predictors in the calibration period (Figure 4). The gases NO$_2$ and CO are shown because the NO$_2$ sensor responds to numerous factors



including other pollutants (i.e., cross-sensitivity) and the CO sensor exhibits a non-linear response to

temperature (Levy Zamora, 2022). The median RMSE of the corrected $PM_{2.5}$ sensor is plotted as a

function of RH and temperature coverage since they have been shown to drive biases in the $PM_{2.5}$

sensor data (Sayahi et al., 2019; Levy Zamora, 2022; Levy Zamora et al., 2018a). If the coverage of

key predictors is high, this indicates that the conditions during the calibration period are representative

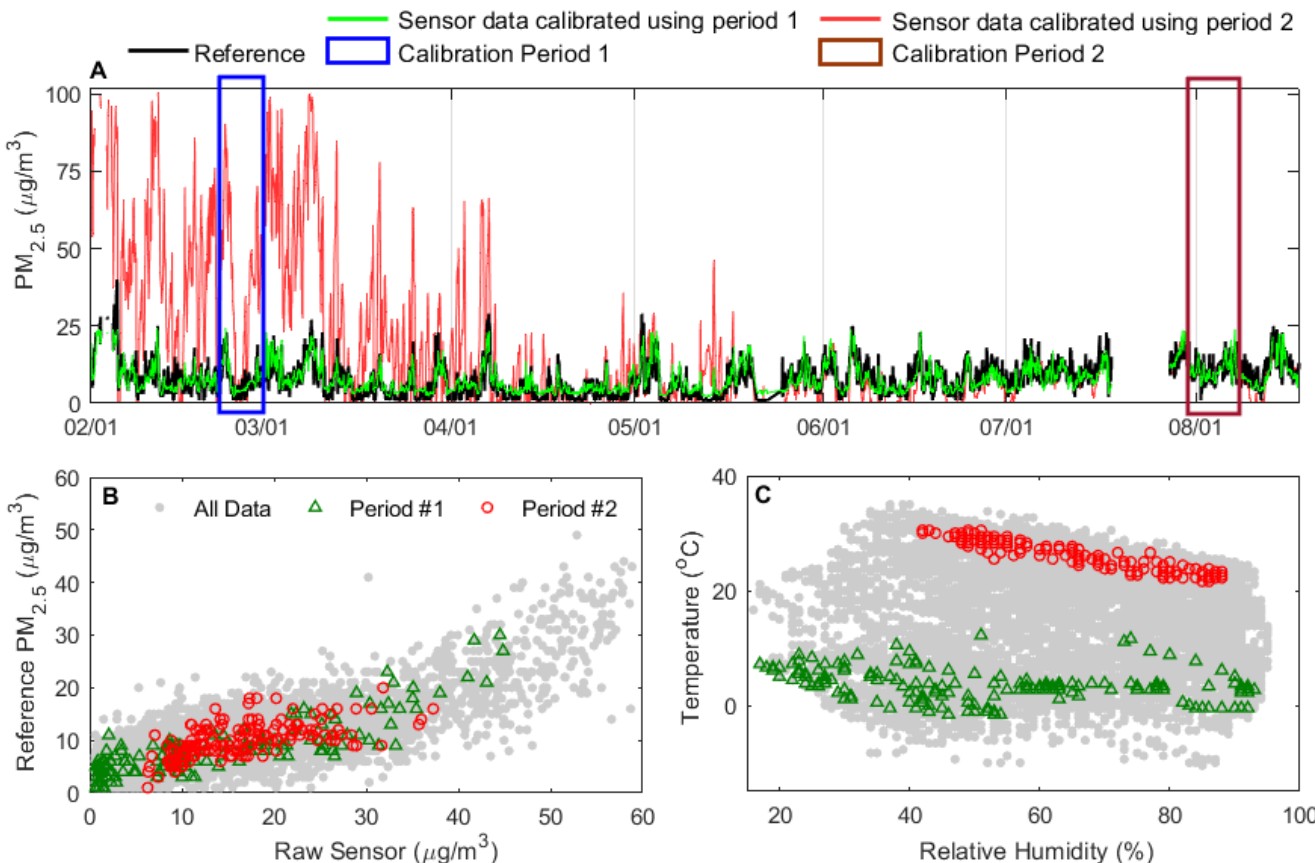

**Figure 3. Example comparison of two potential one-week calibration periods. These were selected to illustrate the range of potential RMSE values that can result from using different periods of the same co-location duration. In the example here, "Calibration Period 1" yielded more accurate concentrations (shown in green; RMSE = 3.1 μg/m³), while "Calibration Period 2" performed poorly when considered across the whole evaluation period (shown in red; RMSE = 19.5 μg/m³). A) The calibrated $PM_{2.5}$ (μg/m³) time series are shown using the two test calibration periods and the reference data (black) from February to August 2019. B) Scatterplot of $PM_{2.5}$ data from the two calibration periods compared to reference data in comparison to the full data set. C) Comparison of RH and ambient temperature for the two calibration periods compared to data from the full year.**



of the evaluation period (i.e., they cover a similar range of values). In general, the calibrations for PM$_{2.5}$ become more accurate (lower RMSEs) as the RH coverage increases (i.e., moving to the right in Figure 4A), and there is a slight improvement with increasing temperature coverage (i.e., Figure 4A moving upwards). The lowest RMSEs were observed when the coverage was high for both temperature and RH. To further clarify the influence of coverage on calibration outcomes, the median RMSEs as a

function of temperature and RH coverages when the PM$_{2.5}$ concentration coverage was greater than 50% are shown in Table 3. RH strongly influences the sensor's raw output, particularly compared to temperature (Levy Zamora et al., 2018b; Levy Zamora, 2022; Sayahi et al., 2019). To yield the best performing calibration outcomes, highly influential cross-sensitives or environmental factors (i.e., RH) should have a minimum coverage of about 70% and secondary factors (i.e., temperature) should have

a minimum coverage of about 50%.

The NO$_2$ sensor exhibits cross-sensitivities to O$_3$ and NO in addition to responding to temperature and RH (Li et al., 2021; Levy Zamora, 2022), so an adequate calibration period should cover an adequate range for all four parameters. The reference NO$_2$ concentrations ranged between 1 and 58 ppb, with a median concentration of 5 ppb. In general, the RMSEs in the NO$_2$ plots decrease as the RH

(Figure 4C x-axis), temperature (Figure 4C y-axis), and O$_3$ coverage increase (Figure 4D x-axis), but the gradient is more clearly seen in the NO coverage (i.e., moving upwards on the y-axis in Figure 4D). The O$_3$ sensor is an example of another sensor that exhibits a cross-sensitivity to another common pollutant (NO$_2$; not shown).

For all three sensors in Figure 4, the RMSEs decreased as the concentration coverage compound

increased, but it was particularly notable for the CO sensor, likely due to the significant differences in seasonal concentrations (e.g., the peak reference CO concentration from December and July were 2950 ppb and 773 ppb, respectively). The reference CO concentrations ranged between 100 and 2950 ppb during the full year, with a median concentration of 199 ppb. This indicates that a period with only





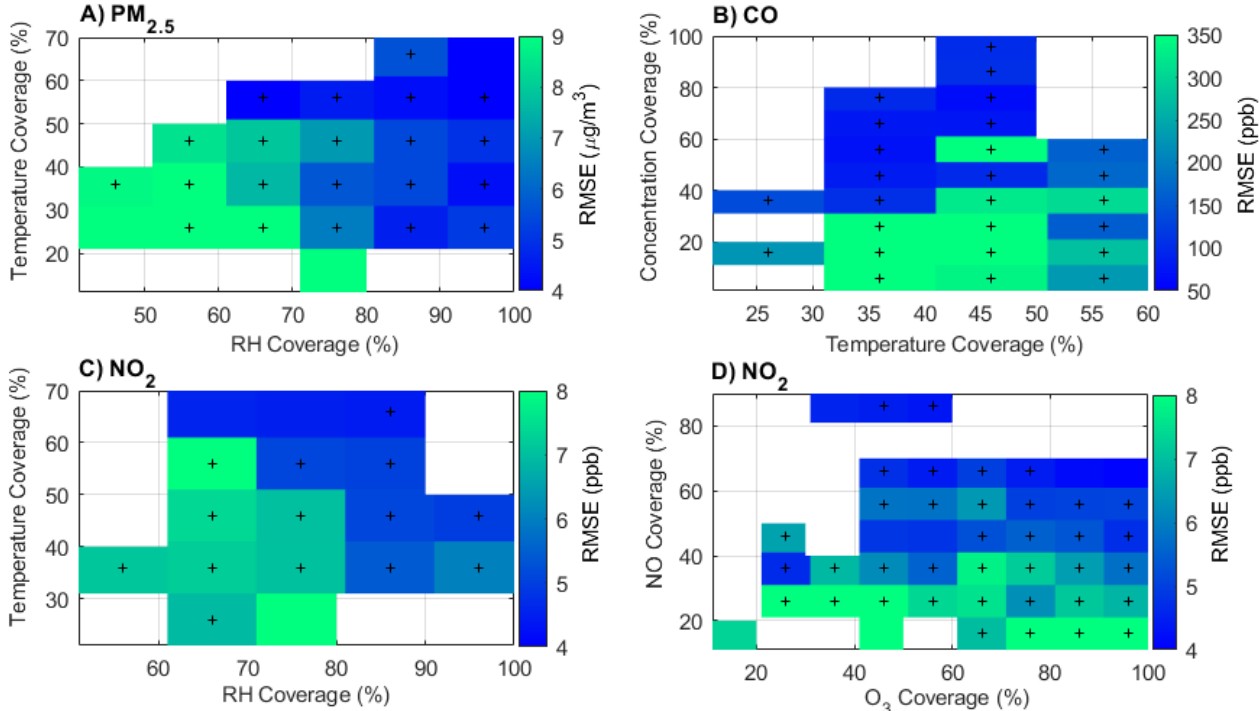

**Figure 4. Median RMSE values for PM₂.₅, CO, and NO₂ sensors are shown as a function of data coverage (i.e., representation) of observed ambient conditions for key predictors within 1-week calibration periods. Bluer colors indicate better calibration results with lower RMSE. The + markers indicate where there were at least 25 calibration runs that fell within that box. The "coverage" values indicate the representativeness of the one-week calibration period compared to the full data set across all seasons. For example, if the temperature ranged from 0 to 40 °C over the full year and a given calibration period ranged from 0 to 12°C, the temperature coverage of that calibration period would be 30% (i.e., Δ12°C/40°C). The ambient temperature and RH ranged between -11 and 36 °C and 14 and 95% over the full year,**

low concentrations may not be able to yield as accurate calibration curves if the evaluation period has

a much broader concentration range than observed during the calibration period. In the CO sensor

panel (Figure 4B), greater temperature coverage generally resulted in lower RMSEs, but a key factor

for the CO sensor is that the calibration must cover warm temperatures if the calibration is going to be

applied to warm seasons. This is due to the notably different responses to high and low temperatures.

This CO sensor exhibits minimal temperature effects below about 15°C but strongly responds to

warmer temperatures (i.e. the sensor will overestimate concentrations at higher temperatures if not

properly calibrated) (Levy Zamora, 2022). More specifically, if a calibration period only included





temperatures below 15°C, that data could not reasonably be extrapolated to a warmer period because it would not be able to correct for this overestimation at high temperatures. Sensors with more linear responses are less sensitive to this issue because a smaller range may be more accurately extrapolated.

We note that the NO and $O_3$ sensors also exhibit non-linear responses to temperature.

It is important to mention that Baltimore, MD is a region that experiences a broad range of meteorological conditions each year, so the co-location duration must be long enough to capture an adequate range of conditions to fully characterize the calibration curves. The pollutants also exhibit significant seasonal variation at this location. In other regions where the weather conditions are less

variable, shorter co-location durations may be more likely to produce accurate results. This is the primary reason why employing a "coverage" approach might be a more useful approach for identifying appropriate co-location durations. Also, we were applying the calibration equations on data from a full year, but shorter co-location durations would likely be satisfactory if the calibration and measurement period were going to be completed under similar conditions (e.g., within one season). For example, if

| PM2.5 RMSE from 1-week calibrations with > 50% concentration coverage | Increasing Temperature Data Coverage ----> | | | | |
|---|---|---|---|---|---|
| | T range > Δ10°C (Coverage >~20%) | T range > Δ15°C (Coverage >~30%) | T range > Δ20°C (Coverage >~40%) | T range > Δ25°C (Coverage >~50%) | T range > Δ30°C (Coverage >~60%) |
| RH range > Δ40% (Coverage >~50%) | 4.7 (3.2 – 17.2; 21%) | 4.7 | 4.7 | 4.4 | 3.8 (3.3 – 11.5; 2%) |
| RH range > Δ48% (Coverage >~60%) | 4.7 | 4.7 | 4.7 | 4.4 | 3.7 |
| RH range > Δ56% (Coverage >~70%) | 4.3 | 4.3 | 4.4 | 4.3 | 3.7 |
| RH range > Δ64% (Coverage >~80%) | 4.3 | 4.3 | 4.3 | 4.1 | 3.7 |
| RH range > Δ72% (Coverage >~90%) | 4.2 (3.2 – 6.8; 7%) | 4.2 | 4.3 | 3.9 | 3.6 (3.2 – 3.7; 1%) |

*(Left vertical axis label: Increasing RH Data Coverage ----V)*

**Table 3. Comparison of the median RMSE (μg/m³) for PM2.5 from 1-week calibration periods with different coverages of temperature and RH conditions. Only calibration periods with more than 50% coverage of the PM2.5 concentration range were included in the table (>50% corresponds to 26 μg/m³ or more in this dataset). For four scenarios (e.g., PM2.5 coverage > 50%, RH Coverage > 50%, T Coverage > 20%), the 1st percentile RMSE, 99th percentile RMSE, and the percentage of calibrations that exhibited all required conditions are shown (1st - 99th percentile; %). For comparison, the median (1st - 99th percentile) of the PM2.5 1-week calibration periods from the full data set (i.e., no coverage requirements) was 6.6 μg/m³**



we limited the calibration and evaluation periods to within June 1 - August 31, 2019 (peak $PM_{2.5} = 25$ $\mu g/m^3$), 70% of one-week co-locations would have an RMSE below 4 $\mu g/m^3$ and an r > 0.6. Similarly, if we limited the calibration and evaluation periods to between November 1, 2019, and February 1, 2020 (peak $PM_{2.5} = 53$ $\mu g/m^3$), 40% of one-week co-locations would have fulfilled these two requirements. Another benefit of strategically identifying co-location needs is that it may permit users

of sensor networks to co-locate each device in the network for shorter periods to get device-specific calibration equations. By ensuring a minimum coverage of key factors for each device co-location period, calibration data between units would likely be more consistent even if the data was collected from different periods. This would be particularly advantageous for sensors types that exhibit notable variability between units.

If little information is known about key predictors at the measurement sites, which is likely at remote locations, it may be possible to use historical meteorological data and general information about pollutant patterns (e.g., emissions and seasonal concentration patterns) to determine a representative range of conditions. Future work should explore whether a combination of multiple, shorter calibration periods in different seasons may produce reasonable calibrations for year-round data sets. However,

in all cases, it is advisable to increase the estimated co-location periods in case of data loss or unusual air quality events and to increase the probability of well-performing calibrations.

## 4.  Conclusions

In this study, we assessed five pairs of co-located reference and low-cost sensor data sets ($PM_{2.5}$, $O_3$, $NO_2$, NO, and CO) to identify key factors that influence the duration of a co-location required to

calibrate low-cost sensors via co-location. We compared the RMSE and correlation coefficients from co-location periods spanning between 1 and 180 days. While longer co-location periods up to several months generally improved the performance of the sensor, the co-location duration was not as predictive of data accuracy as ensuring that the calibration period covered the span of conditions likely



to be encountered during the evaluation period. We determined that many factors could increase the duration of co-location required, including if a sensor responds to environmental factors such as temperature or RH; if the sensor exhibits a cross-sensitivity to another pollutant; if a response is non-linear to any of these factors; and duration of the full deployment (i.e., within a season or spanning multiple seasons). Particular attention must be given to sensors that exhibit a non-linear response if the actual measurement period (e.g., the evaluation period) is going to extend into another season. These results suggest that co-location time can be minimized if selected strategically based on the typical characteristics of a region. The factors that strongly influence the sensor response should have a minimum coverage of about 70% and secondary factors should have a minimum coverage of about 50%.

### Supplemental Materials

Additional figures shown in the supplemental materials include: 1) the start times of 250 randomly selected $PM_{2.5}$ calibration scenarios, 2) the potential range of Pearson correlation coefficients (r) for three low-cost sensors ($NO_2$, $O_3$, and NO) by co-location length, and 3) a zoomed-in comparison of the two potential one-week calibration periods corresponding to Figure 3. This material is available free of charge via the internet at <link>.

### Acknowledgments

This publication was developed under Assistance Agreement No. RD835871 awarded by the U.S. Environmental Protection Agency to Yale University. It has not been formally reviewed by the Environmental Protection Agency (EPA). The views expressed in this document are solely those of the authors and do not necessarily reflect those of the Agency. The EPA does not endorse any products or commercial services mentioned in this publication. The authors thank the Maryland Department of the Environment Air and Radiation Management Administration for allowing us to collocate our



sensors with their instruments at the Baltimore sites. M.L.Z is supported by the National Institute of Environmental Health Sciences of the National Institutes of Health under award numbers K99ES029116 and R00ES029116. The content is solely the responsibility of the authors and does not

necessarily represent the official views of the National Institutes of Health. D.R.G. would also like to acknowledge support from HKF Technology (a Kindwell company). A.D. is supported by the National Science Foundation DMS-1915803 and the National Institute of Environmental Health Sciences (NIEHS) grant R01ES033739. C.B. is supported by the National Science Foundation Graduate Research Fellowship Program under grant no. DGE1752134. Any opinions, findings, conclusions, or

recommendations expressed in this material are those of the author(s) and do not necessarily reflect the views of the National Science Foundation.



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
