# Peer review of "Identifying optimal co-location calibration periods for low-cost sensors"

_EGUsphere, 2022_

## Author Response (AR1)

Overall, the paper is very well written and presents its conclusions clearly. I recommend it for publication following some minor additions and corrections noted below.
The authors want to thank the reviewers for taking the time to provide comments. We have addressed the comments below.

My biggest concern is that using a fixed total amount of sensor data, as the calibration period is increased, the evaluation period is decreased. Comparing results across calibration periods of different lengths could potentially be misleading. Ideally, calibration periods of the same length would be used in all cases; however, this is practically difficult with limited data. A comment to this effect should be added in the paper as a caveat for the presented results.
Response: We acknowledge that this is a constraint caused by a limited data set. Given that we consider several pollutants, there is not one season that captures the full dynamic range of all the sensors. Also, we wanted to evaluate the calibrations consistently across sensor types in as many seasons as was permitted by the available data. To supplement our current analysis, we have added an additional supplemental figure (SF4; See below) referenced in section 3.2 for an analysis of the PM data where the 250 randomly selected calibration periods were from between 02/2019 and 11/2019 and the evaluation period was held to 11/2019-02/2020 for all of the considered calibrations. The results were consistent with the original method.

We have also added to the methods section, "Ideally, evaluation periods of the same length would be used in all cases; however, this is challenging with a limited data set and when comparing pollutants with notably different seasonal trends."

While the use of linear regression approaches to calibration is a reasonable way to approach the analysis, it is by no means the only approach to low-cost sensor calibration. In particular, methods for accounting for the non-linear impacts of various predictors, including quadratic regressions and various machine learning approaches, may be more appropriate. While it is not necessary to exhaustively investigate these here, some mention of these alternative approaches should be made, for example as a topic of future work. Similarly, while using simple "coverage" as a metric to test the appropriateness of the calibration period to the evaluation period is a reasonable first approach, more sophisticated comparisons of the statistical distributions of predictors across these periods could also be applied in future analysis and might also be mentioned here.
Response: We agree linear approaches may not be the best method for all low-cost sensors, but the popularity of linear models is due to their simplicity which makes them accessible to more users. We did investigate simple and interpretable non-linear forms like quadratic and splines with one breakpoint in a previous paper (Evaluating the performance of using low-cost sensors to calibrate for cross-sensitivities in a multipollutant network, https://doi.org/10.1021/acsestengg.1c00367). The calibration models proposed in that work were applied in this manuscript.

We have added discussion on suggested future work: "Future work should evaluate if employing methods that account for any non-linear responses of key predictors can further optimize the calibration of low-cost sensors as well as if more sophisticated comparisons of the statistical distributions of predictors across calibration periods are beneficial."

I would strongly suggest that the datasets used for this analysis be made publicly available if this has not already been done, and the data repository be linked in the paper. This will facilitate other researchers investigating the dataset to determine appropriate calibration strategies for their particular needs.
Response: Search center researchers plan to post data from the network together, including this subset of data. Upon request to the corresponding author, the authors can share the data from this publication

**Specific Comments**
Line 16: "mm" should be micrometers.
Response: This has been corrected.

Line 18: "randomly" should be "randomly selected".

Response: This has been corrected.

Line 80: "was" should be "were".
Response: This has been corrected.

Line 90: What was the increment of the calibration durations? E.g., "ranging from 1 to 180 consecutive days in X day increments". This can be inferred from the presented results, but it is best to explicitly state it as well.
Response: This has been added. "For each hypothetical calibration co-location scenario (i.e., ranging from 1 to 180 consecutive days in 1 day increments), 250 sample calibration test periods were randomly selected of that duration."

Line 115: Please elaborate on what is meant by "time", e.g., hour of the day, day of the week, age of the sensor, etc. Based on later comments I assume it is the age of the sensor, but this should be specified.
Response: In a previous publication, we assessed the change in baseline response over time. Time here refers to the time the data were collected. Therefore, the betas produced clarify how the sensor response changes per unit of time in the calibration period that was not accounted for by the other predictors. We have clarified in the text, "The CO sensor model included temperature, RH, and time, where time refers to the current date and time that the data were collected."

Line 202: "2.5" should be subscripted.
Response: This has been corrected.

Figure 4: For completeness, plots similar to these should be created for all sensors and all predictors and included in the supplemental information.
Response: This has been added in Supplemental Figure 5. "The O3 sensor is an example of another sensor that exhibits a cross-sensitivity to another common pollutant (NO2; not shown in the main text), which has been demonstrated in a previous work (Levy Zamora, 2022). Additional examples of coverage of key variables for all the sensors are shown in Supplemental Figure 5."

Line 269: Remove "compound".
Response: This has been corrected.

Table 3 Caption: The bottom of the caption may be cut off. Also, the "required conditions" should be specified here.
Response: This has been corrected. "Table 3. Comparison of the median RMSE (µg/m3) for PM2.5 from 1-week calibration periods with different coverages of temperature and RH conditions. Only calibration periods with more than 50% coverage of the PM2.5 concentration range were included in the table (>50% corresponds to 26 µg/m3 or more in this dataset). For four scenarios (e.g., PM2.5 coverage > 50%, RH Coverage > 50%, T Coverage > 20%), the 1st percentile RMSE, 99th percentile RMSE, and the percentage of calibrations that exhibited all required conditions (e.g., RH > X % and T > X%) are shown (1st - 99th percentile; %).  For comparison, the median (1st - 99th percentile) of the PM2.5 1-week calibration periods from the full data set (i.e., no coverage requirements) was 6.6 µg/m3 (3.1 – 18.3 µg/m3)."

Line 302: "was" should be "were".
Response: This has been corrected.

Line 311: Remove "and".
Response: This has been corrected.

Line 317-319: Regarding the statement "…the co-location duration was not as predictive of data accuracy…" this might not be entirely supported by your results as you present them, since you do not explicitly perform a meta-analysis of using either duration or coverage as a predictor of performance metrics. You might consider doing such an analysis, or slightly rephrasing this statement.

Response: We have modified the text to state "While longer co-location periods up to several months generally improved the performance of the sensor, optimal calibration could be produced from shorter co-location lengths if the calibration period covered the span of conditions likely to be encountered during the evaluation period."

Line 334: The "<link>" is missing here.
Response: Thank you for noting this. We have modified it to state, "This material is available free of charge via the internet at https://egusphere.copernicus.org/preprints/2022/egusphere-2022-200/egusphere-2022-200-supplement.pdf."

**Supplemental Figure 4.** To supplement our current analysis method where the evaluation period is flexible in order to evaluate more seasons, here we show an analysis of the PM data where the 250 randomly selected calibration periods were from between 02/2019 and 11/2019 and the evaluation period was 11/2019-02/2020 for all of the considered calibrations. The potential range of A) RMSE and B) correlation coefficients (r) for a given co-location length. C) The starting times for each of the 250 calibrations for the one-day analysis are indicated in red, and the evaluation period is shown in gray.

[Figure]

**Supplemental Figure 5.** Additional examples of coverage of key variables for all five sensors using 1-week calibration scenarios. A-C) PM (Temperature, RH, and PM concentration range), D-F) CO (Temperature, RH, and CO concentration range), G-I) $NO_2$ (Temperature, RH, $NO_2$ concentration range, $O_3$ concentration range, and NO concentration range), J-L) NO (Temperature, RH, NO concentration range, and CO concentration range), and M-O) $O_3$ (Temperature, RH, $O_3$ concentration range, and $NO_2$ concentration range). The bluer squares indicate lower RMSE values (more accurate calibrations).

[Figure]

Comments on "Optimizing co-location calibration periods for low-cost sensors" by Zamora et al.,

At the outset, the manuscript was very well drafted, and the analysis was thorough.

The authors want to thank the reviewers for taking the time to provide comments during this busy season. We have addressed the comments below.

Some of my comments are below:

1. The title of the paper is 'optimizing …. calibration periods…..', but I feel that this work hasn't optimized the period, instead it gave suggestions on how to optimize. Authors can think of tweaking the title a bit.
   Response: This has been modified to, " Identifying optimal co-location calibration periods for low-cost sensors"

2. Typo in line 4 of the abstract (mm)
   Response: This has been corrected.

3. Page 3: line numbers 59 and 60, instead of mentioning longer and shorter, please specify the actual duration.
   Response: This has been added. "They reported that longer calibration periods (i.e., six weeks) produced fits with lower bias than fits from shorter calibration periods (i.e., 1 week). In that study, the one-week calibrations yielded the best $R2$ values."

4. Section 2.2: Line 4: what is meant by 'total duration'? Suggest rephrasing lines 2 – 4 for clarity. Currently, the sentence is confusing.
   Response: The sentence has been modified to state, "For each hypothetical calibration co-location scenario (i.e., ranging from 1 to 180 consecutive days in 1 day increments), 250 sample calibration test periods were randomly selected of that duration."

5. Page 5: line 115/116: is the variable 'time' refers to cumulative time or hour of the day?
   Response: In a previous publication, we assessed the change in baseline response over time. Time here refers to the time the data were collected. Therefore, the betas produced clarify how the sensor response changes per unit of time in the calibration period that was not accounted for by the other predictors. We have clarified in the text, "The CO sensor model included temperature, RH, and time, where time refers to the current date and time that the data were collected."

6. Suggest providing mean/summary of pollutant values for the study period in any one of the tables (or as a separate table)
   Response: This has been added to supplemental Table 1.
   Supplemental Table 1. Descriptive statistics of the reference data used in the calibration models from the full year.

|  | Mean | Median | Range |
|---|---|---|---|
| $PM_{2.5}$ ($\mu g/m^3$) | 8.4 | 7 | 1-53 |
| CO (ppb) | 261 | 199 | 100 -2950 |
| $NO_2$ (ppb) | 8.5 | 5.5 | 1-58 |
| $O_3$ (ppb) | 30.1 | 32 | 1-110 |
| NO (ppb) | 3.1 | 0.5 | 0.1-136.5 |

7. Suggest providing NRMSE values in addition to RMSE values to have an idea on how much is the percentage of the error with respect to mean
   Response: We have created an NRMSE table (Supplemental Table 2).

|       | 1 Day | 1 Week | 1 Month | 6 Weeks | 3 Months | 6 Months |
|-------|-------|--------|---------|---------|----------|----------|
| $PM_{2.5}$ | 0.85 (0.13 – 8.11) | 0.12 (0.08 – 0.42) | 0.09 (0.08 – 0.25) | 0.09 (0.08 – 0.18) | 0.09 (0.08 – 0.14) | 0.08 (0.08 – 0.09) |
| CO | 4.21 (0.19 – 48.8) | 0.24 (0.05 – 3.23) | 0.09 (0.04 – 0.35) | 0.07 (0.05 – 0.21) | 0.06 (0.04 – 0.14) | 0.06 (0.05 – 0.08) |
| $NO_2$ | 0.4 (0.11 – 2.27) | 0.15 (0.08 – 0.35) | 0.11 (0.07 – 0.19) | 0.12 (0.07 – 0.15) | 0.11 (0.07 – 0.14) | 0.10 (0.07 – 0.13) |
| $O_3$ | 7.3 (0.14 – 119.59) | 0.55 (0.11 – 2.43) | 0.16 (0.08 – 0.31) | 0.16 (0.08 – 0.24) | 0.17 (0.08 – 0.28) | 0.15 (0.12 – 0.18) |
| NO | 0.12 (0.03 – 5.06) | 0.06 (0.02 – 0.56) | 0.03 (0.02 – 0.06) | 0.03 (0.02 – 0.04) | 0.03 (0.02 – 0.03) | 02 (0.02 – 0.03) |

8. The figure captions can be shortened
   Response: We would prefer to keep the figure captions thorough, so they stand alone for the reader. We will shorten it if necessary to meet journal requirements.

9. Section 3.2: RMSE of 4 and r of 0.6, what are the criteria for these values?
   Response: These two values were selected because most of the satisfactory models exhibited RMSE and r-values within these parameters. However, they were not included as a recommendation for evaluating PM models, but as a way to compare the seasonal differences as described in the paragraph.

---

## Author Response (AR2)

1.  It seems that tables are included as figures #4, Supplemental Figure #5. If it is so, they must be re-labelled as tables and the references in the manuscript text must be adjusted accordingly.

Response:  The figures are correctly labeled as figures in the text.

2. Please ensure that the colour schemes used in your maps and charts allow readers with colour vision deficiencies to correctly interpret your findings. Please check your figures using the Coblis – Color Blindness Simulator (https://www.color-blindness.com/coblis-color-blindness-simulator/) and revise the colour schemes accordingly.

Response: This has been completed.